# Diversity and Composition of Culturable Microorganisms and Their Biodeterioration Potentials in the Sandstone of Beishiku Temple, China

**DOI:** 10.3390/microorganisms11020429

**Published:** 2023-02-08

**Authors:** Yong Zhang, Min Su, Fasi Wu, Ji-Dong Gu, Jie Li, Dongpeng He, Qinglin Guo, Huiping Cui, Qi Zhang, Huyuan Feng

**Affiliations:** 1MOE Key Laboratory of Cell Activities and Stress Adaptations, School of Life Sciences, Lanzhou University, Lanzhou 730000, China; 2National Research Center for Conservation of Ancient Wall Paintings and Earthen Sites, Department of Conservation Research, Dunhuang Academy, Dunhuang 736200, China; 3Gansu Provincial Research Center for Conservation of Dunhuang Cultural Heritage, Dunhuang Academy, Dunhuang 736200, China; 4Environmental Science and Engineering Group, Guangdong Technion—Israel Institute of Technology, Shantou 515063, China; 5Guangdong Provincial Key Laboratory of Materials and Technologies for Energy Conversion, Guangdong Technion—Israel Institute of Technology, Shantou 515063, China

**Keywords:** stone monuments, fungi, bacteria, biodeterioration, pigment, organic acid, biomineralization

## Abstract

Microbial colonization on stone monuments leads to subsequent biodeterioration; determining the microbe diversity, compositions, and metabolic capacities is essential for understanding biodeterioration mechanisms and undertaking heritage management. Here, samples of epilithic biofilm and naturally weathered and exfoliated sandstone particles from different locations at the Beishiku Temple were collected to investigate bacterial and fungal community diversity and structure using a culture-based method. The biodeterioration potential of isolated fungal strains was analyzed in terms of pigmentation, calcite dissolution, organic acids, biomineralization ability, and biocide susceptibility. The results showed that the diversities and communities of bacteria and fungi differed for the different sample types from different locations. The population of culturable microorganisms in biofilm samples was more abundant than that present in the samples exposed to natural weathering. The environmental temperature, relative humidity, and pH were closely related to the variation in and distribution of microbial communities. Fungal biodeterioration tests showed that isolated strains four and five were pigment producers and capable of dissolving carbonates, respectively. Their biomineralization through the precipitation of calcium oxalate and calcite carbonate could be potentially applied as a biotechnology for stone heritage consolidation and the mitigation of weathering for monuments. This study adds to our understanding of culturable microbial communities and the bioprotection potential of fungal biomineralization.

## 1. Introduction

Stone cultural heritage comprises a precious collection of historical objects possessing important cultural, historical, artistic, and scientific value. Most of the stone relics are large and situated outdoors, suffering from serious erosion due to physical, chemical, and biological influences over time. In particular, large, immobile stone relics weather more seriously due to long-term exposure to harsh climates and environmental conditions (natural and anthropogenic) [1,2]. In the natural environment, stones comprise essential ecosystems, providing an ecological niche that various microorganisms occupy; these microorganisms form biofilms that lead to discoloration of and damage to stone materials (biodeterioration) [3]. Microbial biofilms are collections of microbial cells and extracellular polymer substances (EPSs), along with deterioration products and other deposits and debris [4]. It is widely recognized that bacteria, archaea, fungi, algae, and lichen exist on stone relics under different environmental conditions and at different geographical locations [1,5].

Under natural conditions, freshly cut stone is first colonized by pioneer microbes, such as phototrophic cyanobacteria, algae, and lichen. They can form a thick layer, a patina, and a crust that modify the properties of the stone surface, significantly contributing to the biodeterioration of the stone substrate and inducing its subsequent colonization by heterotrophic bacteria and fungi [2]. Biodeterioration is a slow process caused by the colonizing microorganisms forming a biofilm on the stone surface, which appears in different colors and eventually results in damage [6,7,8,9]. Specific groups (e.g., sulfate-reducing bacteria, nitrifying bacteria, sulfur-oxidizing bacteria) of colonizing microorganisms have been revealed to contribute to the biochemical reactions on the stone, allowing a better understanding of the processes involved, as well as prevention, control, and management [10,11]. To control biodeterioration, antimicrobials (e.g., biocides) have been widely used in the protection of stone cultural heritage [12], but their use to control biodeterioration is controversial and should be employed sparingly.

Among the colonizing microbes, the biological activities of bacteria and fungi can release cations from minerals and precipitate them as the secondary minerals calcium oxalate (CaC_2_O_4_) or calcium carbonate (CaCO_3_) on stone surfaces. The processes of mineral formation through oxidation or reduction of metals that are catalyzed and induced by microorganisms are called “biomineralization” [13,14,15]. Fungal hyphae and secreted oxalic acid are related to oxalates, such as calcium oxalate, which includes whewellite (CaC_2_O_4_·H_2_O) and weddellite (CaC_2_O_4_·2H_2_O) [16]. Calcium oxalate can also be transformed into calcium carbonate by some fungi, meaning that fungi may contribute to calcium carbonate precipitation [17,18,19,20]. Furthermore, fungi can synthesize CaCO_3_ with the two crystal structures of either calcite or vaterite directly. The secondary minerals may protect the stone from environmental damage due to their relative insolubility, but they also cause aesthetic alterations and the deterioration of the stone surface [16,21].

The Beishiku Temple (BT), which was founded in the Northern Wei Dynasty (509 AD), is one of the most well-known Buddhist holy sites on the ancient Silk Road in the eastern Gansu Province of China. Long-term exposure of the site to sunlight, rain, wind, and biological activity in the open-air environment has caused severe and irreversible changes and damage to its physical structure, chemical composition, and aesthetic characteristics. Bioweathering is a major threat to the preservation of such outdoor sandstone monuments at this site, and emergency management is necessary for sustainable conservation. Increasing attention has been paid to investigations of microbial communities using high-throughput sequencing (HTS), salt damage, and mineralogical analysis at this site lately [22,23,24,25,26]. By using both culture-dependent and culture-independent methods, a rich understanding of stone biodeterioration has been developed, making it possible to significantly advance knowledge on the compositions and metabolisms of epilithic microflora and specific biochemical processes [2]. Given the significance of culturable microorganisms in the biodegradation of stone cultural relics, it is critical to determine which microorganisms grow on the stone and what role they play. As a result, this study’s objectives were to: (1) determine the composition of culturable microbial communities and their potential functions; (2) compare culturable microbial communities and their relationships with environmental factors; and (3) test biomineralization processes involving culturable fungi. These findings will facilitate a basic understanding of native microflora and their potential for sandstone deterioration, as well as the opportunities for sustainable protection of stone heritage with biotechnologies.

## 2. Materials and Methods

### 2.1. Site Description

The Beishiku Temple (BT) is located west of the loess plateau in Qingyang, Gansu Province, northwest China (Appendix A), and to the east of Fuzhong Mountain at the confluence of the rivers Pu and Ru, at the coordinates 35°32′00″ N and 137°36′35″ E and an altitude of 1064–1083 m. The temple was excavated from Mesozoic Cretaceous yellow sandstone and currently has 308 caves and 2129 stone statues. The cliff is about 20 m high, facing westward, and 1030 m above sea level. BT is situated in a mild-temperate, sub-humid climate zone with an annual average temperature of 9.9 °C, a relative humidity of 63%, and precipitation of 516.5 mm.

The most common and visible changes at BT include cracking, water seepage and infiltration, damping, weathering, salt efflorescence, color alteration, exfoliation, moss cover, lichen, and microbial colonization and deterioration.

### 2.2. Sampling Strategy

A total of 24 samples were collected from the surfaces of the stone sculptures at BT in April 2018. Outside the caves, two locations were selected for sampling: three exposed caves (numbers 10, 19, and 270) near the ground and one exposed cave (number 281) in the top environment. Inside the caves, only one location was selected for sampling in one semi-open cave (number 9) near ground level (Figure 1; Appendix A). The samples included two types (1—biofilm; 2—weathered powder) and were further divided into six groups: samples from outside the caves at different heights (exposed outside samples near the ground, which were labeled EO, including EO-1 and EO-2, and exposed outside samples from the top environment, which were labeled ET, including ET-1 and ET-2) and weathered samples from inside the caves (semi-open environment samples, which were labeled SO, including SO-1 and SO-2). In this study, various sample types collected from different heights were used to verify the spatial variation and the microflora specificity at different locations.

The surface materials were sampled aseptically using common sampling procedures at the specific locations where the bas-reliefs and sculptures were suffering severely from visible deterioration. Biofilm samples and weathered particulate samples with a small amount of sandstone on the top face were collected. Samples with a total weight of approximately 1.5 g were carefully collected with a sterile scalpel, stored in a 2 mL sterile Eppendorf tube, and placed in an icebox for immediate storage and transport. After collection, the samples were sent to the laboratory for further analysis within 48 h.

### 2.3. FESEM-EDS and XRD Analyses

The surface morphologies and element compositions of the different sample types and crystal grains were investigated using field emission scanning electron microscopy (FESEM, FEI Quanta 450 FEG, FEI, Hillsboro, OR, USA) combined with energy-dispersive spectroscopy (EDS). Each sample was about 1 mm thick and was dried in a germ-free state for 2–3 h. These samples were subsequently fixed onto two-sided conductive adhesive carbon tape and gold spray-coated for 60 s. The working parameters for the SEM-EDS were as follows: 15 KV accelerating voltage, 25 mm electro focusing, 1 Na current probe, and 60–300 s spectral collection [26]. The crystal structures of the samples were characterized from X-ray diffraction analysis (XRD, Rigaku D/Max-2400, Rigaku, Tokyo, Japan) of the powdered samples using a 40 kV accelerating voltage in the range of 10–80° 2θ at a scan rate of 5° min^−1^.

### 2.4. Soluble Salts Analysis

The moisture content (Mc) was determined using the weight loss method after oven drying (24 h at 105 °C) with 0.5 g of sample. The dried samples were homogenized into powder using a mortar and pestle, then mixed with 15 mL of ultrapure water four times in a row to obtain a soluble salt solution via ultrasonic extraction. The above filtrate was used to determine water-soluble cations (Na^+^, NH_4_^+^, K^+^, Mg^2+^, and Ca^2+^) and anions (Cl^−^, NO_2_^−^, NO_3_^−^, and SO_4_^2−^) via ion chromatography (Dionex 600, Dionex, Sunnyvale, CA, USA). The electrical conductivity (Ec) of the sample filtrate was determined using a conductivity meter (Leici DDSJ-318, Leici, Shanghai, China), and the pH value was determined with a pH meter (Sartorius PB-10, Sartorius, Goettingen, Germany).

### 2.5. Environmental Data Acquisition

An iButton^®^ temperature/humidity logger (DS1923, Maxim Integrated, Sunnyvale, CA, USA) was used as measuring equipment to obtain environmental data at a fixed depth below the stone surface (5 mm depth) at different sampling locations. The temperature (Tem, °C) and relative humidity (RH, %) data were recorded hourly for a year using an iButton^®^ inside cave 9 and outside caves 10 and 281. Following statistical analysis, the monthly average values around the sampling were used.

### 2.6. Microbial Enumeration and Identification

The microbes (bacteria and fungi) were isolated using the dilution plate method. A total of 100 mg of sample was dispersed in 900 μL of ddH_2_O in a 2 mL sterile Eppendorf tube and vortexed for 20 min. After evenly mixing the sample suspension, it was diluted in a continuous gradient of 10^−1^–10^−5^. Then, the 100 μL sample dilution was evenly applied to R2A and PDA solid medium agar plates (representing an oligotrophic environment on the stone surface) with three replicates. The Petri dishes were incubated at room temperature for 7–10 days in the dark. Colony-forming units (CFUs) on the R2A (Difco) and PDA (Solarbio) agar plates were enumerated and bacterial and fungal populations were expressed as CFUs per gram of sandstone (CFUs/g). CFUs/g were determined as: C(CFUs/g) = T × B/(0.1 g/1 mL × D mL). In the equations, C is the concentration of culturable bacteria and fungi; T is the total number of colonies on the R2A and PDA agar plates; B is the dilution ratio; and D is the dilution amount for the plate coating.

After enumerating the colonies on the plates, fungal isolates were preliminary classified based on morphological characteristics, including shape, size, color, etc. Subsequently, R2A and PDA agar plates were employed to isolate and purify single colonies using streak culture with different phenotypes, and the pure culture isolates were obtained before DNA extraction and molecular identification.

The genomic DNA of the pure culture isolates was extracted using a commercially available extraction kit according to the manufacturer’s protocols for bacteria (DP302) and fungi (HP Fungal DNA Kit, Omega 3195-01) (Tiangen Co., Beijing, China).

The bacterial 16S rRNA gene was amplified with universal primer pairs (27F/1492R; 5′-AGAGTTTGATCCTGGCTCAG-3′/5′-CTACGGCTACCTTGTTACGA-3′), and the fungal ITS region was amplified using universal primers (ITS1/ITS4; 5′-TCCGTAGGTGAACCTGCGG-3′/5′-TCCTCCGCTTATTGATATGC-3′). The PCR system (25 µL) was composed of 2.5 μL of 10 × Taq Buffer, 2.5 μL of dNTP mixture (2.5 mM), 2 μL of each primer (5 μM), 3 μL of DNA template, 0.4 μL of DNA polymerase (TransGen Biotech, Beijing, China), and 14.6 μL of sterile water to make up the volume.

PCR was performed with an ABI GeneAmp^®^ 9700 Cycler (ThermoFisher Scientific, Waltham, MA, USA), and the bacterial parameters were set as follows: initial denaturation at 94 °C for 3 min; 35 cycles × 94 °C for 1 min, 55 °C for 1 min, and 72 °C for 1.5 min; and a final extension at 72 °C for 15 min. The fungal parameters were set as follows: initial denaturation at 94 °C for 5 min; 35 cycles × 94 °C for 30 s, 55 °C for 30 s, and 72 °C for 1 min; and a final extension at 72 °C for 15 min. PCR products were detected with 1% agarose gel electrophoresis and checked.

The specificity of PCR products was determined using restriction fragment length polymorphism (RFLP) analysis. The PCR products were digested with double-restriction endonucleases. Bacterial enzyme digestion was applied for 3.5 h at 37 °C using *Bsu*RI and *Csp*6I. Fungal enzyme digestion was applied for 4 h at 37 °C using *Bsu*RI and *Hin*fI. The spectral pattern on 2.5% agarose gel was used to separate the digested fragments into clusters.

The PCR products were purified with purification kits (Tiangen) and then cloned with the pGEM-T Vector System (Tiangen Co., Beijing, China) overnight at 4 °C. After ligation, they were transformed into *E. Coli* DH5α-competent cells for blue-white spot screening. White clone spots were moved with a sterilized toothpick into LB liquid containing ampicillin (100 mg/L) under sterile conditions and then shaken overnight at 150 rpm and 37 °C. The pGEM-T vector primer pairs (T7/SP6) were used to examine positive clones via PCR in the same amplification program. The verified cultures were sequenced by a commercial service at TSINGKE Biological Technology (Xi’an, China).

The taxonomic resolution of the ITS partial sequences used in this study had some limitations, meeting the requirements for the description of fungal diversity but not always directly meeting the requirements for identification at the species level. As a result, species names of fungal strains were labeled with the abbreviation “cfr.”, which comes from a Latin term (*conferatum*) meaning “compare to”. This abbreviation is used in taxonomy when a strain is close to a species but other elements are necessary to prove it beyond any doubt.

### 2.7. Evaluation of Biodeterioration Potential of Fungal Strains

#### 2.7.1. Preparation of Spore Suspensions

A 1.0 × 10^5^ CFU/mL fungal spore suspension was obtained with hemocytometer measurements after washing the acid-producing fungi-containing PDA plates with 0.9% sterile saline and 1% Tween 20 [27].

#### 2.7.2. Pigment Generation of Fungal Strains

Czapek-Dox minimal medium was used to inoculate and culture the isolated fungal strains. The medium was composed of 2 g sodium nitrate, 1 g dipotassium phosphate, 0.5 g magnesium sulphate heptahydrate, 0.5 g potassium chloride, 0.01 g iron (II) sulphate heptahydrate, 10 g glucose, 20 g agar, and 1000 mL deionized water and was sterilized at 115 °C for 25 min (the pH was adjusted to 5.5 with 1 mol/L HCl) [28]. The fungal spore suspension (10 μL) was inoculated on Petri plates and incubated at room temperature for 14 days, which was accompanied by recording of pigment production with photographs.

#### 2.7.3. The Acid-Producing Properties of Fungal Strains

The acid-producing strains were screened on CaCO_3_ glucose agar plates. The medium consisted of 5 g calcium carbonate, 10 g glucose, 15 g agar, and 1000 mL deionized water (the pH was adjusted to 8 with 1 mol/L HCl) and was sterilized at 121 °C for 15 min [27]. Fungal spore suspension (20 μL) was inoculated on the Petri plates and incubated at 25 °C for 20 days to evaluate the calcite dissolution ability.

#### 2.7.4. The Measurement of the Impact of Fungal Strains on pH

Czapek-Dox liquid medium has a stable pH value of approximately 7, which was adjusted with 1 mol/L HCl. Fungal spore suspension (10 μL) was added into the liquid medium, and it was shaken (150 rpm/min) at room temperature for 5 days to detect pH changes and for 7 days to measure weight loss in the sandstone blocks.

#### 2.7.5. Antimicrobial Test

Fungal spore suspension (200 μL) was uniformly coated on PDA plates, and then nine common biocides (names with CAS numbers: dichlorophen (9-1, DDM, 97-23-4); 1,2-benzisothiazolin-3-one (9-2, BIT, 2634-33-5); didecyldimethylammonium chloride (9-3, DDAC, 7173-51-5); benzalkonium chloride (9-4, BAC, 8001-54-5); isothiazolinone (9-5, 26172-55-4); 2-octyl-2H-isothiazol-3-one (9-6, OIT, 26530-20-1); natamycin (9-7, 7681-93-8); 2-butyl-1,2-benzisothiazolin-3-one (9-8, BBIT, 4299-07-4); 4,5-dichloro-2-octyl-isothiazolone (9-9, DCOIT, 64359-81-5)) were selected for a subsequent test. The stock solution was prepared with deionized water or ethanol.

The appropriate working concentration was selected from a gradient dilution range of 10^−1^ to 10^−6^, and the dilution with the best inhibitory effect was used for the following experiment. The K-B method was used for the inhibition zone test. First, 4 mm round filter paper was dried after sterilization. Then, the filter paper was soaked in biocides of working concentrations for 1 h and put into a test plate to determine the inhibition zone (the mean value was obtained from three replicates in each plate) after culturing for 24 h. A Vernier caliper was used to measure the inhibition zone with the cross method, and the diameter represented the size.

#### 2.7.6. Analysis of Organic Acid Production

The production of organic acids was investigated with high-performance liquid chromatography (HPLC). A total of 2 mL of the selected fungal culture medium was centrifuged at 1000 rcf/min for 10 min and then filtered through nitrocellulose filters with a 0.22 μm pore size and eluted with 0.01 mol/L NaH_2_PO_4_ at a pH value of 2.65. The HPLC system used was an Agilent 1260 system (Agilent Technologies, Santa Clara, CA, USA) equipped with a G1315D Diode Array Detector (DAD) of 210 nm. Chromatographic separation was performed with a ZORBAX SB-Aq analytical column (250 mm × 4.6 mm i.d., 5 μm, Agilent Technologies, Santa Clara, CA, USA), and the temperature was set at 25 °C. The mobile phase consisted of 97% NaH_2_PO_4_ (0.01 mol/L, pH 2.65) and 3% methanol, with a flow velocity of 1 mL min^−1^. The sample injection volume was 20 μL, with three replicates. The different peaks separated in the samples were used to identify the acid types by comparing them with external standards (oxalic acid, malic acid, lactic acid, acetic acid, citric acid, succinic acid, and fumaric acid). The liquid culture was used as a control.

### 2.8. Fungal Mineralization Characteristics

The fungal strains capable of precipitating carbonate were screened on modified B4 (MB4) medium, which consisted of 2.5 g calcium acetate, 4 g yeast extract, 10 g glucose, 18 g agar, and 1000 mL deionized water (the pH value was unadjusted) [21]. Fungal spore suspension (100 μL) was inoculated onto sterile MB4 plates. After 7 days of incubation at 25 °C in the dark, morphological observations of crystal formation were recorded using an optical microscope and SEM. The fungal strains were incubated in liquid MB4 medium at 25 °C on a rotary shaker (150 rpm/min) for 7 days, with the pH of the solution measured every 24 h. The plates with the attached crystals were dried at 65 °C, the further morphologies and elemental compositions were investigated using FESEM combined with EDS, and crystal structures were analyzed using XRD.

### 2.9. Statistics

Routine enumeration was applied for the counting of microbial colonies. Plates were divided into equal sectors (from halves up to eighths) for counting. Plate CFU counts were estimated by multiplying one sector count by the total number of sectors. The charts for the CFU population, diversity, community composition, and XRD were plotted in Origin 2021 (OriginLab Corporation, Northampton, MA, USA). Redundancy analysis (RDA) was undertaken in Canoco v5.0 (Microcomputer Power, Ithaca, NY, USA). A neighbor-joining tree was reconstructed using MEGA software v7.0 (Available online: http://www.megasoftware.net, accessed on 15 October 2020). The functional taxa of the cultural bacteria were annotated with the python script (collapse_table.py) in the Python 3 environment by FAPROTAX database (version 1.2.2, Available online: https://pages.uoregon.edu/slouca/LoucaLab/archive/FAPROTAX/lib/php/index.php, accessed on 7 September 2022). The functional guilds of the cultural fungi were analyzed using the python script (FUNGuild.py) in the Python 3 environment by FUNGuild database (Online version, Available online: http://www.funguild.org, accessed on 7 September 2022). All representative sequences were submitted to GenBank with bacterial accession numbers MT076216-MT076258 and OL854196-OL854199 and fungal accession numbers MN982315-MN982349.

## 3. Results

### 3.1. SEM Analysis

The micromorphology of the collected biofilms and sandstone samples was examined using SEM analysis. A massive number of microbial filaments were observed in the biofilm samples, and they were tightly attached to or even penetrated the particles of the stone substrate (Figure 2a,b). The weathered sandstone showed an abundant quantity of clastic particulates (Figure 2c,d). However, there were no visible mycelia interacting with the substrate samples of the weathered particulates. Salt crystal efflorescence was observed in the weathered stone samples (Figure 2e,f).

### 3.2. Physiochemical Properties of Sandstone Samples

The ion chromatography analysis results showed high concentrations of K^+^, Mg^2+^, Ca^2+^, Cl^−^, SO_4_^2−^, NO_3_^−^, and Na^+^ in the stone samples, with Ca^2+^ being the most abundant cation at more than 3 mg/g (Appendix A). Cl^−^, SO_4_^2−^, NO_3_^−^, and Na^+^ concentrations varied significantly between samples and were especially high in sample EO-2, which had higher electrical conductivity. ET-1 had significantly higher moisture content than the others, but biofilm samples (EO-1, SO-1, ET-1) tended to have higher moisture content than weathered samples (EO-2, SO-2, ET-2). The pH of biofilm sample SO-1 was significantly higher than that of weathered sample SO-2, both of which were collected from inside the caves.

### 3.3. Populations and Diversity of Culturable Microorganisms

The isolated bacterial and fungal strains were characterized in terms of their various morphologies and colors and their hyphal appearance and pigment secretion. Some of the representative isolates of the bacteria and fungi are shown in Appendix A.

The average culturable bacterial populations in the different samples ranged from 4.51 ± 0.35 × 10^5^ CFUs g^−1^ (SO-2) to 4.5 ± 0.52 × 10^7^ CFUs g^−1^ (ET-1). The populations for EO-1 and ET-1 were significantly higher than the others (Figure 3a). The average culturable fungal populations ranged from 2.44 ± 0.3 × 10^3^ CFUs g^−1^ (SO-2) to 3.56 ± 1.03 × 10^5^ CFUs g^−1^ (ET-1). The microbial population for ET-1 was significantly higher than the others (Figure 3b). Furthermore, most of the microbial populations in the biofilm samples (EO-1, SO-1, ET-1) were much higher than those in weathered samples (EO-2, SO-2, ET-2).

The Shannon–Wiener diversity index values for the bacteria, which revealed no significant differences between samples, were in the following order: ET-1, ET-2 > EO-1, EO-2 > SO-1, SO-2 (Figure 3c). The fungal diversities for EO-1, EO-2, and ET-2 were significantly higher (*p* < 0.05) than that of SO-2, and the order was: EO-1, EO-2 > ET-2, ET-1 > SO-1, SO-2 (Figure 3d). As a result, the diversity of the culturable bacteria and fungi outside the caves was greater than that inside.

### 3.4. Community Structures and Composition of Microorganisms

All bacterial and fungal isolates were PCR-amplified for the 16S rRNA gene and internal transcribed spacer (ITS) regions, respectively, and the sequences were analyzed and grouped. Following the NCBI BLAST tool in the nucleotide database, all the bacterial sequences (Appendix A) were classified into 39 different species, with 28 genera and 6 phyla affiliated with them. Fungal sequences (Appendix A) were classified into 25 different species, with 14 genera and 2 phyla affiliated with them. The neighbor-joining tree for these sequences is shown in Figure 4.

In each sample group, Actinobacteria and Proteobacteria predominated; their summed relative abundances ranged from 37.49% (SO-1) to 91.13% (EO-1). Bacteroidetes only accounted for a high abundance in SO-1 (26.30%) and SO-2 (8.30%). Firmicutes had a high abundance in EO-1 only (6.05%). However, the collectively contributing proportions of the remaining two phyla (Gemmatimonadetes and Deinococcus-Thermus) varied from 0% (EO-1 and ET-1) to 4.76% (EO-2) (Figure 5a).

Bacterial community composition and diversity varied among the different groups. Samples EO-1 (17 genera) and EO-2 (19 genera) contained most of the genera and were dominated by *Arthrobacter* (22.06%), *Sphingomonas* (20.71%), and *Nocardioides* (9.65%). The dominant genera in ET-1 (13 genera) and ET-2 (14 genera) were *Streptomyces* (4.31%) and *Blastococcus* (3.15%), respectively. SO-1 (ten genera) and SO-2 (ten genera) contained fewer genera and were dominated by *Flavobacterium* (25.66%), *Sphingomonas* (25.14%), and *Arthrobacter* (7.72%), and *Flavobacterium* (7.20%), respectively (Figure 5b).

The fungal community composition varied greatly among the different samples. Samples EO-1 (11 genera) and EO-2 (10 genera), collected near the ground level from outside the caves, had the most abundant genera and were dominated by *Penicillium* (30.28%) and *Engyodontium* (13.76%). These were followed by ET-1 (five genera) and ET-2 (four genera), collected outside from the top of Beishiku Temple; *Epicoccum* (10.09%) and *Alternaria* (18.35%) were the dominant genera. SO-1 (three genera) and SO-2 (two genera), from inside the caves, had the minimum number of genera, and the dominant genus was *Cladosporium*, with abundances of 50.04% and 15.60%, respectively (Figure 5c). The results generally agree with the differences in the Shannon–Wiener diversity index values at the different sites (Figure 3c,d).

### 3.5. Factors Influencing Microbial Community

According to the RDA analysis, the community structures of the bacteria and fungi varied among the different sample groups, and environmental factors had a great influence on the distribution of community structures. pH (17.8%), temperature (11.7%), Cl^−^ (10%), and Ca^2+^ (9.8%) were the factors that contributed the most. The key factor for the bacterial community explained 11% of the variation in pH (*p* < 0.05) (Figure 6a). For the fungal community, pH (22.9%), temperature (21.7%), relative humidity (14.3%), Ca^2+^ (9.6%), and moisture content (9.5%) played important roles. pH and temperature, as the main environmental factors, explained 16.2% and 15.4% of the variation in the fungal community, respectively (*p* < 0.05) (Figure 6b).

### 3.6. Predicted Ecological Functions

The prediction of the ecological functions of the bacterial and fungal communities is displayed in Figure 7. A large amount of the bacterial sequence was assigned to chemoheterotrophy or aerobic chemoheterotrophy, and they were observed equally in all sample types (72.7%). The functions human_pathogens_all and human_associated were more prevalent in samples taken from outside of the caves (EO-1, SO-2, ET-1, and ET-2). The functions related to the nitrite or nitrate biogeochemical cycle were predicted; they displayed higher relative proportions in the weathered samples outside the caves (EO-2 and ET-2) (Figure 7a).

The predicted functional characteristics of the fungi demonstrated a diverse situation. A total of 11 fungal guilds were identified. The main fungal guilds found across all samples were in the following order: wood saprotroph, undefined saprotroph, animal pathogen, plant pathogen, and dung saprotroph. The five functional guilds accounted for substantial relative proportions in all sample types, except for samples from inside the caves (SO-1 and SO-2), in which only one guild (animal pathogen) showed a higher relative proportion (Figure 7b).

### 3.7. Biodeterioration and Biocide Susceptibility Test

All isolated fungal strains were tested for pigment production and calcite dissolution capability (Appendix A). The results showed that four of the fungal strains (*Ophiobolus* cfr. *artemisiae*, *Chaetomium* cfr. *luteum*, *Epicoccum* cfr. *nigrum*, and *Penicillium* cfr. *chrysogenum*) could secrete visually observable organic pigments (Figure 8a–d). Five of the fungal strains (*Penicillium* cfr. *chrysogenum*, *Cladosporium* cfr. *sphaerospermum*, *Penicillium* sp., *Penicillium* cfr. *citrinum*, and *Penicillium* cfr. *rubens*) showed calcite dissolution capability, with clearing zones around the colonies on the agar plates (Figure 8e–j). This suggests that the activity of these fungi causes the deterioration of selective inorganic components of sandstone.

Next, nine biocides potentially usable for cultural heritage protection were selected for the biocide susceptibility testing of the isolated fungi. The mean inhibition zones were 11.8 mm, 12 mm, 8.75 mm, 8.45 mm, 9.3 mm, 12.65 mm, 9.35 mm, 9.2 mm, and 11.4 mm, which showed that 2-octyl-2H-isothiazol-3-one (OIT) had the best inhibitory effect; its average inhibition zone size was the largest (Appendix A).

### 3.8. Production of Organic Acids and Deteriorative Potentials

The acidification of the fungi with calcite dissolution abilities was measured in a liquid medium, and the value of the pH decreased significantly after 7 days of culturing (Appendix A), mostly due to the secreting of organic acids, which were determined with HPLC. According to the HPLC results, all five fungal isolates could produce oxalic acid and lactic acid, and some were found to produce acetic acid, succinic acid, and fumaric acid (Table 1).

The fungal isolates capable of acid production were then used for simulated biodeterioration assessment with sandstone test blocks collected from the same sandstone around Beishiku Temple. Weight loss was observed for all stone blocks inoculated with acid-producing fungi when compared to control groups that did not receive fungi inoculation. There was a significant difference in weight loss between the test samples for *P.* cfr. *citrinum* (*p* < 0.01) and *C.* cfr. *sphaerospermum* (*p* < 0.01) and the control samples (Appendix A).

### 3.9. Biomineralization Characteristics

Our results showed that the five acid-producing strains were capable of precipitating distinctive crystals in an MB4 medium. Optical microscopy revealed that the mineral crystals formed around fungal mycelia had different shapes and sizes (Appendix A). SEM analysis revealed typical crystals with various morphological forms, including tetragonal bipyramids (Figure 9b,g,j) and eight-faced bipyramids (Figure 9a,c,e,f,i). Interestingly, these crystals were mainly observed at the bottom of the culture medium.

The pH value of the culture media inoculated with fungi capable of mineralization decreased below 7, except for that of *P.* cfr. *citrinum* (Appendix A). This suggested that the acids produced by the fungal strains were utilized, thus resulting in no significant drop in pH values. The results of the XRD analysis further supported the finding that the main crystal structures precipitated by the five acid-producing strains were mainly composed of weddellite (CaC_2_O_4_·2H_2_O) and calcite crystal (CaCO_3_) (Figure 10).

## 4. Discussion

This study was the first to investigate the community diversity, structure, and biodeterioration capabilities of culturable microorganisms on stone monuments at Beishiku Temple, combining the findings with an analysis of environmental factors primarily related to the bacterial and fungal community structures. This may contribute to a better understanding of and provide more comprehensive information on the microbial ecology and biodeterioration mechanisms of Beishiku sandstone.

### 4.1. The Characteristics of Bacterial Community Structure

Bacteria are significant surface-colonizing microflora that compromise the structural integrity of stone cultural heritage. The bacterial composition for the different sampling locations and sample types varied. Actinobacteria and Proteobacteria were the most abundant culturable bacterial phyla, followed by Bacteroidetes, Firmicutes, Gemmatimonadetes, and Deinococcus-thermus. This was consistent with our previous HTS analysis [24,25], as well as reports by others on stone heritage sites [29,30,31].

Actinobacteria, Proteobacteria, and Bacteroidetes are all involved in hyphal and biofilm formation. Firmicutes, the members of which are generally very resistant to heat and drought and widely found in soil and arid environments, accounted for only 4.6% of all organisms [32]. In addition, two uncommon bacterial phyla, Gemmatimonadetes (2.6%) and Deinococcus-thermus (0.4%), were also isolated, both containing only one genus (*Gemmatimonas* and *Deinococcus*). Previous results for the biofilm community at the Bayon Temple in Angkor showed the presence of Deinococcus-thermus [33], which are resistant to irradiation, desiccation, and high temperatures, especially the genus *Deinococcus*.

Actinobacteria were mainly dominated by *Arthrobacter* and *Microbacterium*, which are the two most widely reported heterotrophic bacterial genera affecting cultural heritage. They belong to a common group of lithotrophic microorganisms with nutritional diversity and a marked oligotrophic capacity that enables them to withstand extreme environments, including the harsh conditions of stone surfaces, temperature fluctuations, and high salinity [29]. *Microbacterium* species are common airborne microorganisms and can be deposited anywhere.

Proteobacteria were mainly dominated by *Sphingomonas*, which are subordinate to Alphaproteobacteria, accounting for 56.84% of the phylum.

Previous research has suggested that Alphaproteobacteria are the most metabolically active group [34] and play a dominant role in limestone biofilm [35]. *Sphingomonas* have been found in a variety of habitats, including soil and water, as they can use organic compounds as substrates and grow in nutrient-poor environments. They can also produce a yellow pigment to protect themselves from UV rays. These properties are conducive to the survival of microbes in adverse environments.

The predicted chemoheterotrophy and aerobic chemoheterotrophy functions in all sample types were consistent with the functional predictions based on our previous HTS analysis. The predicted functions associated with nitrite or nitrate transformation in the weathered samples outside the caves (EO-2 and ET-2) corresponded with the predicted nitrification process functions from the HTS analysis (Figure 7a) [24]. The functional prediction taxa of the culturable bacteria could be matched with HTS data; these taxa were rich in all sample groups and indicated the dominance of heterotrophic bacteria. Notably, human-related functions were predicted to be dominant outside the caves; this may have been a result of tourist activities.

Nearly all isolated bacterial genera were found in our latest analysis [24] but with variable abundance. Only one oligotrophic medium was selected for this study, which resulted in some apparent limitations affecting the indigenous microflora information due to the conventional culture methods.

### 4.2. The Characteristics of Fungal Community Structure

The three fungal phyla Ascomycota, Basidiomycota, and Zygomycota were detected in this study, with Ascomycota being overwhelmingly dominant, which was very similar to our previous findings from sandstone stelae [25]. The abundant genera *Cladosporium*, *Penicillium*, *Engyodontium*, and *Alternaria* contributed more than 10% of the mycoflora in this study, accounting for 73.62% of the total CFUs, and they are always the most common fungi detected at cultural heritage sites [36,37].

However, the fungal taxa varied among the sampling sites and were affected by several factors. Under moderate or humid environmental conditions, fungal communities are usually dominated by filamentous fungi, including *Cladosporium*, *Alternaria*, and *Epicoccum* [38]. Sample groups SO-1 and SO-2 from inside the caves, which had high relative humidity levels of 73.5% and 89.72%, respectively, were dominated by *Cladosporium* in this semi-open environment (Appendix A). In contrast, sample groups ET-1 and ET-2 from outside the caves at a height of 10.5 m above the ground (Appendix A), which had a saturated relative humidity of 101.32% and high relative humidity of 82.73%, respectively, were dominated by the genera *Epicoccum* and *Alternaria* (Figure 5c) due to the continuous direct infiltration of rainwater from the main body of the mountain.

Similarly, the genera *Alternaria*, *Epicoccum*, *Cladosporium*, and *Phoma* showed relatively high abundance and were among the top 50 genera in weathered samples in the HTS results of our latest study [26]. The taxa of the airborne fungi that could cause stone biopitting and the corresponding functional guilds (Figure 7b) were wood saprotroph, undefined saprotroph, animal pathogen, plant pathogen, dung saprotroph, and saprophytes. This was much like the previously predicted functions from the HTS analysis, especially the high relative proportion of the animal pathogen functional guild inside the caves [26]. Interestingly, bat guano and traces of bat activity were found inside the sampled caves during the fieldwork, and this coincided with the high proportion of the predicted animal pathogen functional guild inside caves. It also suggests that the caves may be potential reservoirs of pathogenic microorganisms. These findings confirm the dominance of these filamentous fungi and their potentially harmful effects in stone cultural heritage conservation and protection.

### 4.3. Effects of Environmental Factors

The greatest population of culturable bacteria and fungi appeared in sample type ET-1 (Figure 3), which was influenced by the high relative humidity and temperature (Appendix A). The culturable population of fungal spores generally increased with temperature and humidity [39]. Meanwhile, the biofilm samples (EO-1, SO-1, ET-1) generally contained higher populations than the weathered ones (EO-2, SO-2, ET-2), likely due to the biofilms’ organically rich ability to retain water from the atmosphere [40].

The microbial community diversity in stone cultural relics in dry climates is greater than that in wet ones [41], as shown by the highest diversity index for the bacteria in this study (Figure 3c). The bacteria’s characteristics may aid in their colonization of the sandstone substrate and correspond to the surrounding environment and conditions in the top environment of the cliff (ET-1, ET-2), where they are exposed to the open environment, including direct sunlight. There were no significant differences in the bacterial Shannon–Wiener index values in this study (Figure 3c) because bacterial biofilm communities tend to be more stable on stone [42].

For the weathered particulate samples, soluble salts were enriched on the stone surface and salt crystallization was observed (Figure 2c,d), which is not conducive to microbial growth. The RDA analysis showed that pH, temperature, and relative humidity were the main factors determining the structures of the microbial community (Figure 6). As shown by previous studies, temperature and relative humidity are the main environmental factors affecting microorganisms under different environmental conditions [36,43]. To overcome the limitations of sampling from stone monuments, a greater number of samples from each position could provide more accurate statistical data and results. Water or moisture is essential for the growth and survival of microorganisms on stone and in building environments, and the available water for microflora may originate from an invisible source; e.g., humidity in the atmosphere when a thermal gradient is available that allows condensation to take place on the surface [44]. High temperatures and humidity provide favorable environmental conditions for microbial growth and dissemination, increasing the diversity and abundance of microorganisms. With regard to pH values, microorganisms usually grow within a certain pH range, and near-neutral pH conditions favor the uptake of mineral nutrients among the majority of microorganisms [45].

### 4.4. Fungal Biodeterioration and Fungal Control

Microbial interactions with stone monuments have been extensively investigated. Microorganisms are capable of producing pigments that can cause aesthetic damage to stone monuments. The production of biopigments is a defense mechanism in response to adverse environmental conditions, such as high UV radiation, desiccation, and hypersalinity, and it is affected by environmental conditions, such as relative humidity, temperature, and nutrient availability, in various culturing media [46,47]. As a result, different behavior may be observed in lab analyses and actual conditions. Production of organic acid [48,49] and inorganic acid [50,51], as well as exopolymeric substances (EPSs) [52], also contributes to physical and chemical deterioration, with specific mechanisms and biochemical reactions involved [41,53]. Four of the isolated strains could produce pigments: *O.* cfr. *artemisiae*, *C.* cfr. *luteum*, *E.* cfr. *nigrum*, and *P.* cfr. *chrysogenum*. Fungi from the genera *Aspergillus*, *Chaetomium*, *Cladosporium*, *Epicoccum*, *Gibberella*, and *Penicillium* have been shown to produce pigments in biodegradative potential tests [27,54,55]. In this study, the fungal genus *Ophiobolus* was found to be able to produce pigments for the first time.

It is well-known that filamentous fungi produce low-molecular-weight organic acids [55,56,57]. Organic acids can reduce pH values and interact directly with the substratum materials of cultural heritage [58]. An assessment of the acid-producing properties of fungi isolated from cultural heritage materials showed that many species of the genera *Aspergillus*, *Cladosporium*, and *Penicillium* are effective acid producers [27,36,55], as was also found for *Rhizopus* [59]. However, production of organic acids by the isolated strain *Rhizopus* was not detected in our study, probably due to the characteristics of the strain itself or the physiological conditions and metabolic stage.

Biocide sensitivity testing of fungal isolates from stone monuments can identify the most resistant species or the most promising biocides [60]. Here, nine biocides commonly used in cultural heritage protection were selected for the biocide susceptibility test of the isolated fungi (Appendix A). The difference in the efficacy of the biocides was small, but OIT presented the best inhibitory effects against all the isolated fungal strains. OIT is a modern, broad-spectrum, effective biocide with a low application dose and good compatibility, and it has been applied for the protection of various stone cultural relics, including the sandstone of a Roman archaeological site in Italy [61] and the granite stonework of the Monastery of San Martiño Pinario in Spain [62]. It could be a promising fungicide suitable for the emergency control of biodeterioration at our study site.

### 4.5. Fungal Biomineralization and Potential Applications

Biomineralization is a ubiquitous phenomenon in which microorganisms facilitate mineral formation. Fungi are directly involved in the biogeochemical transformation of elements on a global scale by excreting organic acids [63,64]. Fungal biomineralization is primarily attributed to the production of organic acids, such as acetic, citric, gluconic, fumaric, glyoxylic, and oxalic acids, which precipitate with various metals or metalloids through the salt formation and complexation process [48,49,57].

Biomineralization of stone relics has been widely reported. Fungi react with mineral constituents of stone substrates to form weddellite, and strains from the genera *Penicillium*, *Cladosporium*, *Chaetomium*, and *Aspergillus* all have mineralization potential [56,65,66,67,68,69]. The strains with biomineralization capabilities identified here were those from the genera *Penicillium* and *Cladosporium*. The existence of crystal phases of calcium oxalate (weddellite) and calcium carbonate (calcite) was supported by XRD analysis (Figure 9). Such large amounts of secreted acids and attached biomineralization products are direct evidence of heritage deterioration, which may lead to mechanical damage (chalking, peeling, and fissures) and aesthetic alterations (pigmentation) in the stone substratum [20,70,71,72].

The precipitation of calcium oxalate and calcium carbonate is influenced by fungal growth, as well as environmental conditions. Therefore, it is necessary to explain the precipitation process for calcium oxalate and calcium carbonate to provide a solid basis for subsequent microbial control and enable its use as a biotechnological tool for heritage protection [73,74].

Five of the isolated acid-producing strains were capable of precipitating weddellite and/or calcite, and weddellite precipitation probably resulted from the reaction between oxalic acid and calcium acetate [67]: Ca^2+^ + C_2_O_4_^2-^ + nH_2_O = Ca(C_2_O_4_)·nH_2_O↓. For calcite precipitation, biomineralization may have occurred in regions of Ca^2+^ accumulation, and carbon dioxide produced by oxalate oxidation and fungal respiration can increase CO_3_^2−^ concentration in the local environment, which is conducive to calcite precipitation [75]: Ca^2+^ + CO_2_ + 2OH^−^ → CaCO_3_↓ + H_2_O.

Several of the findings suggested that calcium oxalate biominerals are critical to the formation of crusts on rock surfaces and that calcium oxalate can be transformed into calcium carbonate by oxalic acid-producing (oxalotrophic) microorganisms. The pathway whereby oxalate is transformed into carbonate is induced by oxalotrophic fungi through calcite precipitation, which enhances stone cementation and healing [76]. Carbonates are common biominerals, and numerous studies have shown that fungi, as one major group among others, are involved in calcite formation.

Microbiologically induced carbonate precipitation (MICP) has attracted extensive attention as an appropriate and innovative means for soil stabilization, sandstone cementation, and repair of concrete in environmental and civil bioengineering. Biogenic carbonate formation is a very effective and feasible protective measure for the consolidation of macroporous stones [77], and this technology has been proposed for infrastructure maintenance and repair. Fungal systems may serve as a promising bioprotection strategy for further development and implementation. Moreover, Ca^2+^ is the most abundant cation in ecosystems (>3 mg/g; Appendix A) and has the mineralogical characteristics of calcite. Considering the existence of calcium carbonate, the biomineralization mechanism induced by fungi may provide a deeper insight into the biotechnology of stone monuments.

## 5. Conclusions

This study investigated the culturable microorganisms on the sandstone of Beishiku Temple and their mineralization potentials. The microbial community composition and its relationships with environmental factors were further analyzed. The physiological and biochemical characteristics of fungi involved in biodeterioration were explored simultaneously. The dominant bacterial genera were *Sphingomonas* and *Arthrobacter*, followed by *Flavobacterium*, *Roseomonas*, *Pseudomonas*, and *Nocardioides*, while the dominant fungal genus was *Cladosporium*, followed by *Penicillium*, *Engyodontium*, and *Alternaria*. The culturable bacteria’s main predicted functions were chemoheterotrophy and aerobic chemoheterotrophy, and those of the culturable fungi were saprotroph and pathogens. Interestingly, animal pathogens were predominant in samples inside the caves, which was consistent with our previous HTS results. The most promising biocide candidate for biodeterioration control at this study site was 2-octyl-2H-isothiazol-3-one (OIT).

Among the culturable fungi, several strains were capable of producing pigments, and some of the strains producing acids showed biomineralization potential through the precipitation of weddellite and calcite. In samples collected from various types of locations with various appearances and microenvironmental conditions, the compositions of the culturable microbial communities varied greatly. Temperature, RH, and pH value were the factors closely associated with the distributions of the microbial community and composition. This research provides scientific evidence regarding culturable microflora and support for the sustainable management and conservation of the endangered sandstone heritage at this study site.

## Figures and Tables

**Figure 1 microorganisms-11-00429-f001:**
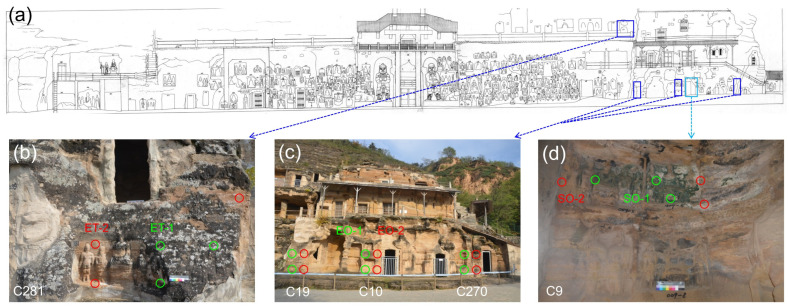
Relative positions of sampling sites at Beishiku Temple. Green circles refer to biofilm samples; red circles refer to weathered particulate samples. (**a**) is the sampling site and (**b**–**d**) are zoomed portions of subfigure (**a**) from where the samples were collected.

**Figure 2 microorganisms-11-00429-f002:**
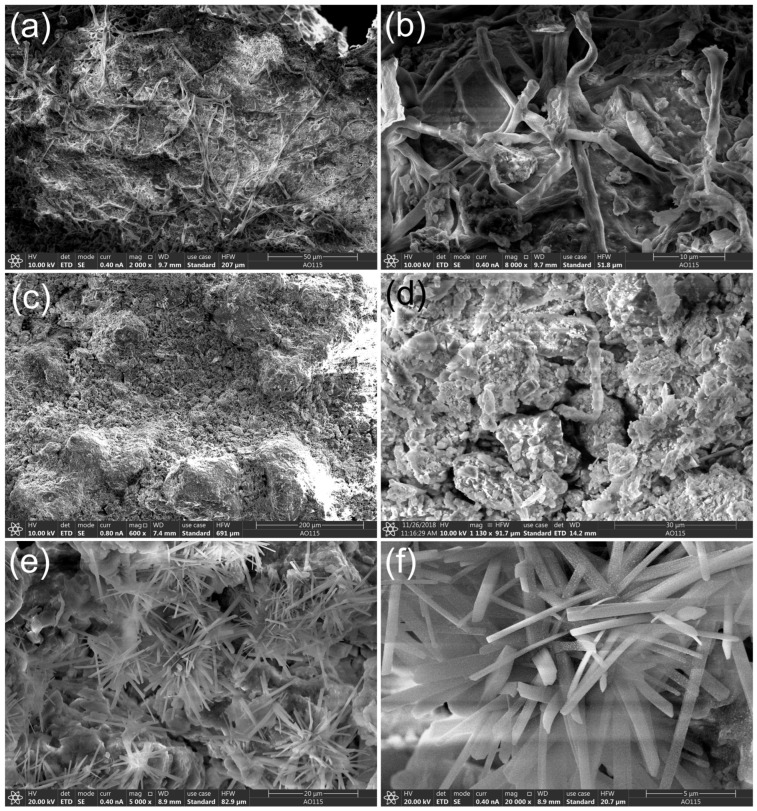
Morphological structures of sandstone samples from Beishiku Temple. Scanning electron micrographs (SEM) of microbial hyphal colonization (**a**,**b**), weathered sandstone (**c**,**d**), and salt crystal efflorescence (**e**,**f**).

**Figure 3 microorganisms-11-00429-f003:**
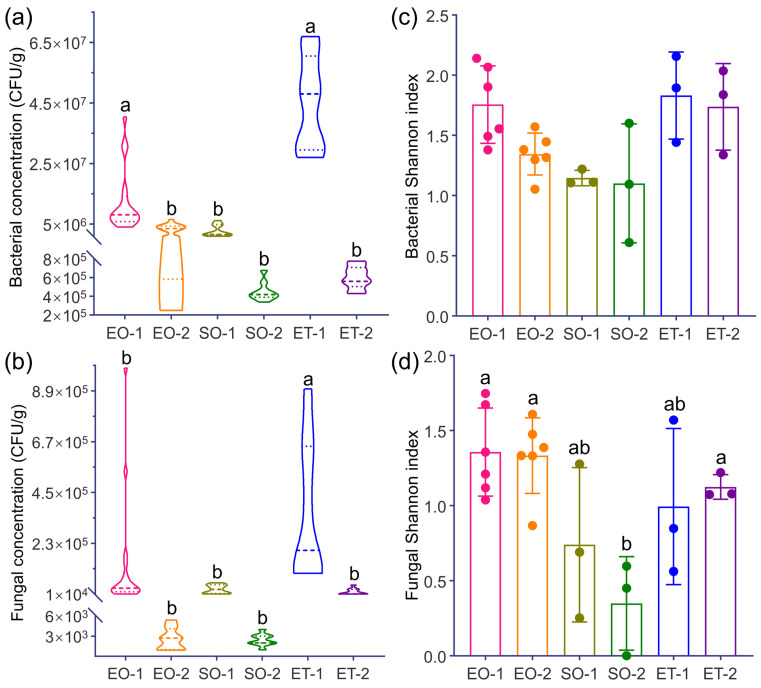
The concentrations and Shannon index values for culturable bacteria and fungi (CFUs g^−1^) from the different sampling sites. Significant results are indicated with different letters. (**a**) Bacterial concentration; (**b**) Fungal concentration; (**c**) Bacterial Shannon index; (**d**) Fungal Shannon index.

**Figure 4 microorganisms-11-00429-f004:**
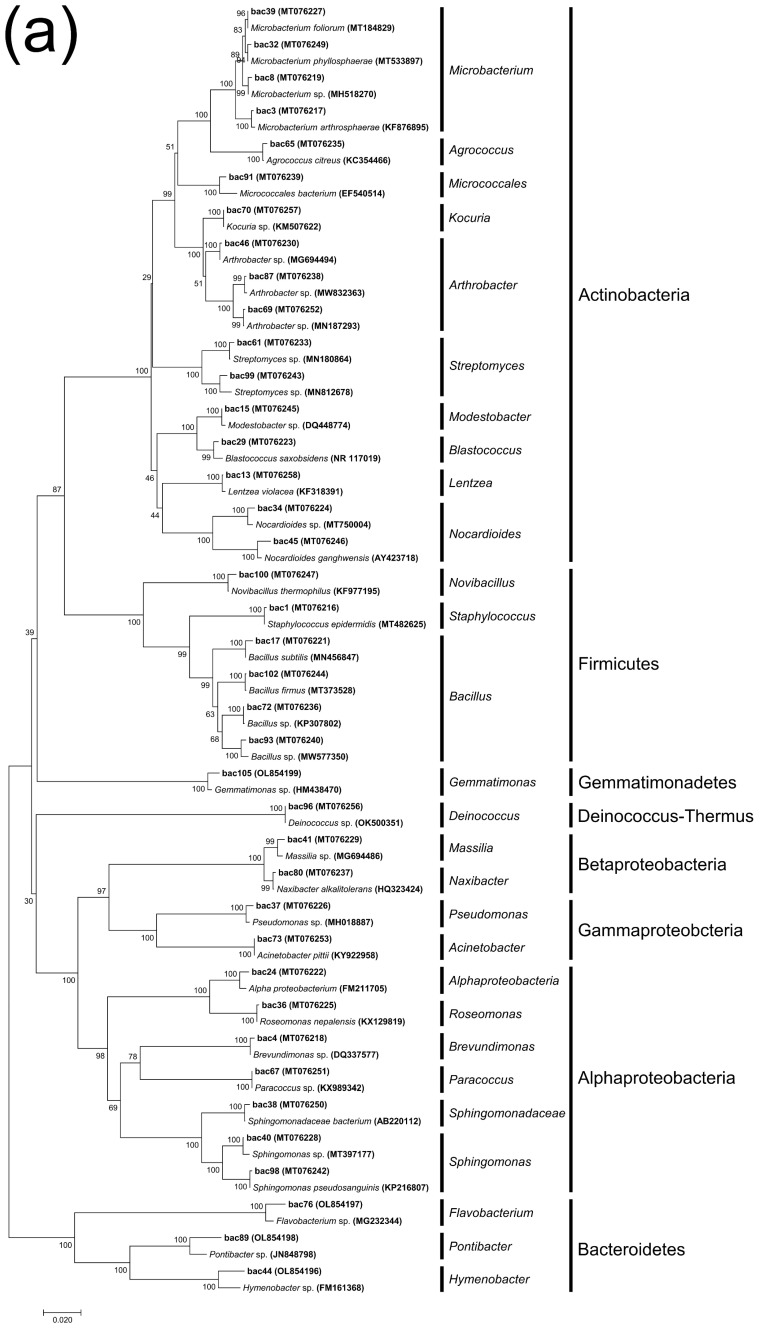
Neighbor-joining tree representing strains of bacteria (**a**) and fungi (**b**) isolated from samples based on the 16S rRNA gene sequences and the ITS gene sequences, respectively.

**Figure 5 microorganisms-11-00429-f005:**
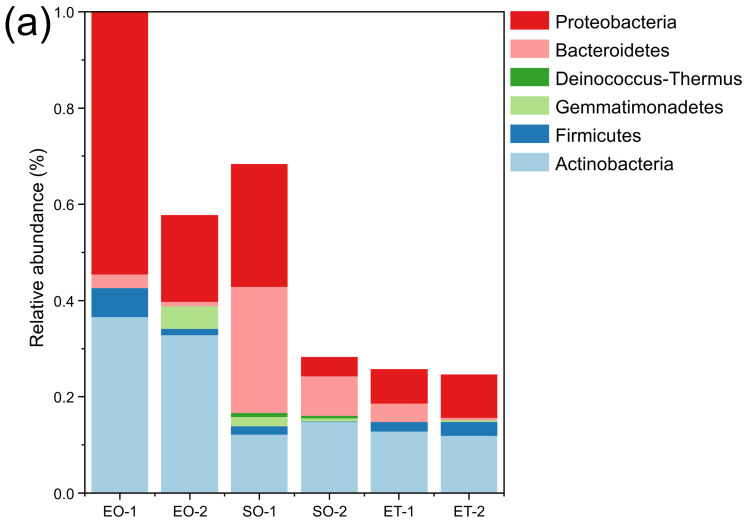
Relative abundance of the culturable bacteria at the phylum level (**a**) and genus level (**b**) and the culturable fungi at the genus level (**c**).

**Figure 6 microorganisms-11-00429-f006:**
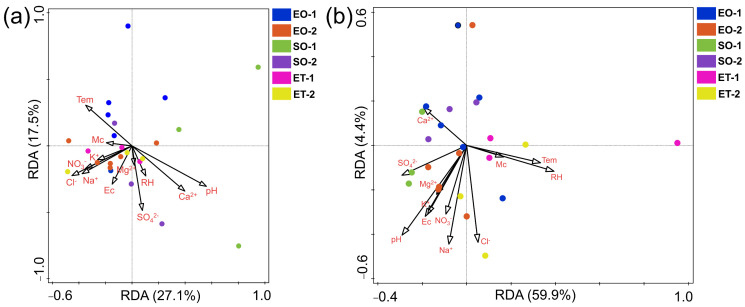
Redundancy analysis of bacterial (**a**) and fungal (**b**) communities and environmental parameters from the different sites.

**Figure 7 microorganisms-11-00429-f007:**
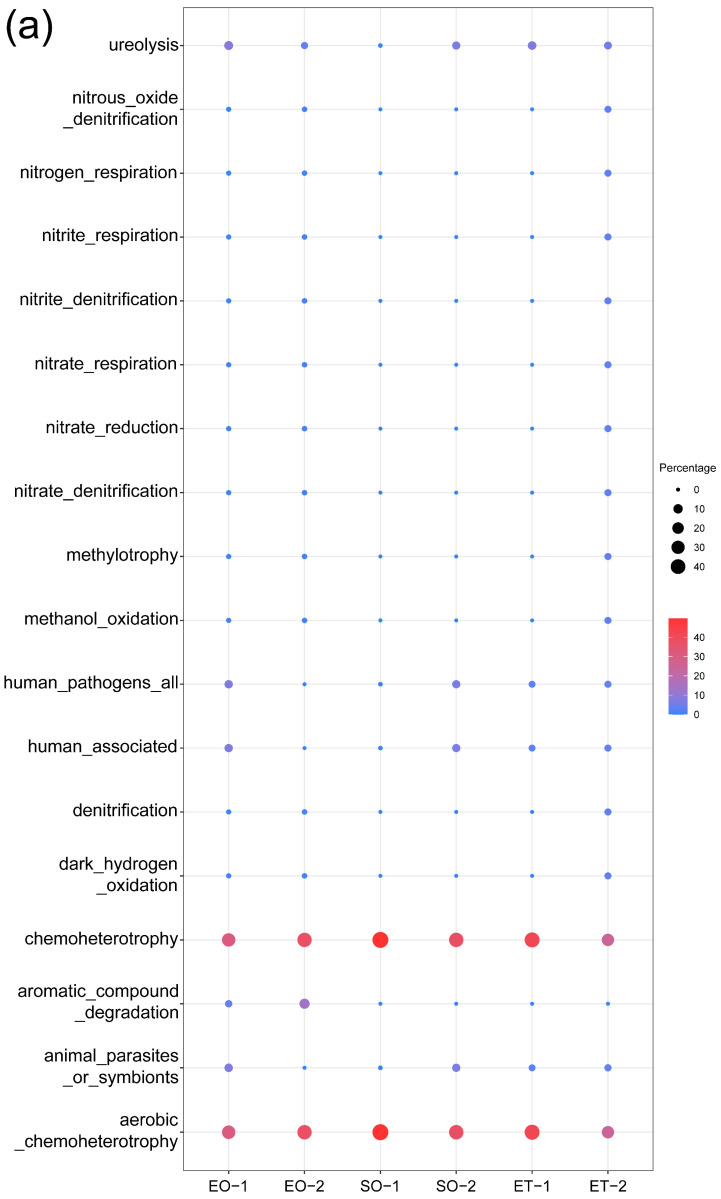
Prediction of the ecological functions for (**a**) bacteria and (**b**) fungi.

**Figure 8 microorganisms-11-00429-f008:**
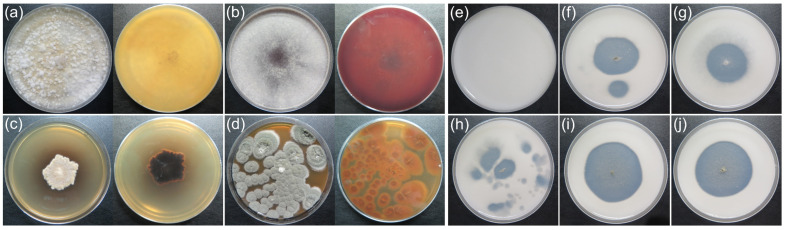
Fungal pigment secretion (day 15, Czapek-Dox) and the phenomenon of calcite dissolution (day 20, CaCO_3_ glucose agar) in a medium. (**a**) *O.* cfr. *artemisiae*; (**b**) *C.* cfr. *luteum*; (**c**) *E.* cfr. *nigrum*; (**d**,**f**) *P.* cfr. *chrysogenum*; (**e**) transparent non-inoculated plate; (**g**) *C.* cfr. *sphaerospermum*; (**h**) *Penicillium* sp.; (**i**) *P.* cfr. *citrinum*; (**j**) *P.* cfr. *rubens*.

**Figure 9 microorganisms-11-00429-f009:**
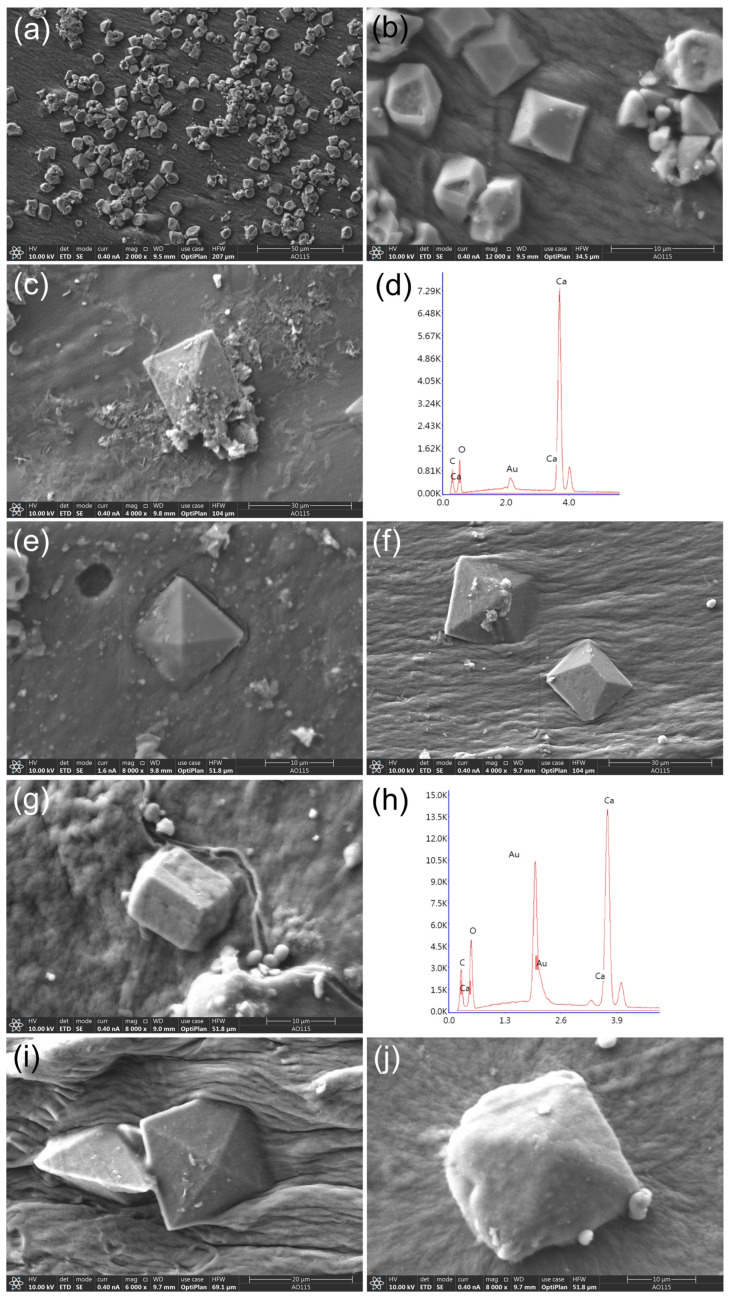
SEM analysis of the crystals precipitated by acid-producing strains (day 7, MB4 medium). (**a**,**b**) *P.* cfr. *rubens*; (**c**) *P.* cfr. *chrysogenum*; (**d**) EDS spectrum corresponding to (**c**); (**e**) *P.* cfr. *citrinum*; (**f**,**g**) *C.* cfr. *sphaerospermum*; (**h**) EDS spectrum corresponding to (**g**); (**i**,**j**) *Penicillium* sp. (**b**,**g**,**j**) Calcite crystals (CaCO_3_); (**a**,**c**,**e**,**f**,**i**) weddellite (CaC_2_O_4_·2H_2_O).

**Figure 10 microorganisms-11-00429-f010:**
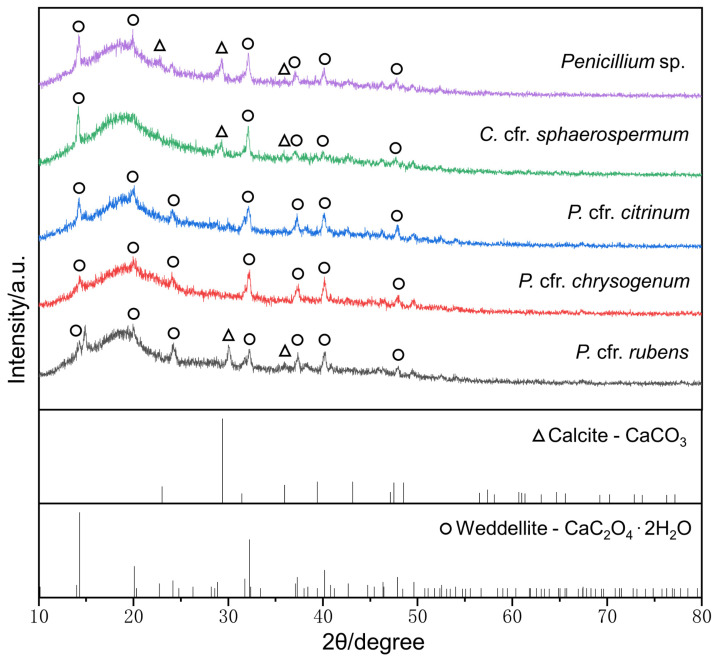
XRD analysis of MB4 medium inoculated with acid-producing strains. The XRD standard spectra were plotted from the PDF standard cards (calcite—PDF#05-0586, weddellite—PDF#17-0541).

**Table 1 microorganisms-11-00429-t001:** Secretion profiles of different types of organic acids for selected fungal strains.

Fungal Strain	Organic Acid							
	Acetic Acid	Citric Acid	Fumaric Acid	Lactic Acid	Malic Acid	Oxalic Acid	Pyruvic Acid	Succinic Acid
*P.* cfr. *rubens*	−	−	+	+	−	+	−	+
*P.* cfr. *chrysogenum*	−	−	+	+	−	+	−	+
*P.* cfr. *citrinum*	−	−	−	+	−	+	−	+
*C.* cfr. *sphaerospermum*	+	−	+	+	−	+	−	−
*Penicillium* sp.	−	−	−	+	−	+	−	+

## Data Availability

Not applicable.

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
