# Peer review of "Diversity and Composition of Culturable Microorganisms and Their Biodeterioration Potentials in the Sandstone of Beishiku Temple, China"

_microorganisms, 2023, doi:10.3390/microorganisms11020429_

Round 1
Reviewer 1 Report
Comments:
The paper ”Diversity and composition of culturable microorganisms and their biodeterioration potentials on sandstone Beishiku Temple, China” by Zhang et al. is very interesting, sound, comprehensive and focuses on a case study of a sandstone temple in China. The paper complements one previous one already published (based on HTS analysis performed in the same site).
The paper is very clear, very well written and well designed and organised. The study is complete in my opinion.
The abstract is clear, pointing out the main results.
The Introduction is very well written, but the objectives need to be more clearly stated, pointing out why this study is innovative and important, at least for this monument (see last paragraph of the Introduction and rephrase it accordingly). The authors only refer to that in the discussion, and I think it is important to address it at the end of the introduction.
In this section, and all over the paper I would change “colonized microorganisms” by “colonizing microorganisms” (e.g. lines 58 and 60 and 66).
In line 60, please specify what is the “specific group of microorganisms”. Please state the groups or explain what you mean by specific group…
Line 70. Change “Fungi hyphae” by “Fungal hyphae”
Lines 70-73: please add some more references about this particular production of this mineral by fungi, in several types of stone.
Methods are appropriate and are well explained. The used techniques are adequate to fully respond to the aims of the study.
Line 219. Please begin the sentence by “Two hundred” instead of “200”.
Results are well and extensively described with the appropriate number of tables and figures and supplementary material, clearly presented and are important to the state of the art in this type of monuments. The figures/images are of good quality and elucidative.
In the Results the authors clearly describe the bacterial and fungal compositions, explore the relations between these compositions with the environmental factors, and the biodeterioration capabilities were properly evaluated.
Discussion of the results is also complete and addresses the most important questions.
Just some minor comments: in section 4.4, just one reference #37 (line 558) is not enough to corroborate what is said by the authors. I would add a couple of references more.
The same applies to line 586 and 589 and in line 598 (section 4.5). Please add some more references that can illustrate also these phenomena.
For example, these phenomena have also been described in several types of limestone deteriorated by fungi.
The conclusions are clear and well supported. I like very much the second and last paragraph of the conclusions.
The list of references is satisfactory, well formatted! But I would add some more references as I said above.
I would complete the names of all the authors in reference 33, instead of putting “et al”. It is the only one with “et al.” and there is only one more author in this particular reference…I know that the journal can accept both formats when there are more than 10 authors…In this particular case there are 11 co-authors.
Congratulations for the nice work!
Reviewer 2 Report
The work entitled “Diversity and composition of culturable microorganisms and their biodeterioration potentials on sandstone Beishiku Temple, China” presents a study about the culturable microbial community diversity and their biodeterioration capabilities on stone monuments at Beishiku Temple, in Cina. This is a useful study to increase the knowledge about the microbial ecology and biodeterioration mechanisms of Beishiku sandstone and It is an interesting study approach to apply also in other contexts. The paper is well-organized and structured. Every aspect reported is investigated in deep and well discuss. The negative aspect is the excessive use of supplementary materials (seven figures). These should be added materials that the reader can or not use. However, I think that some figures could be instead to be helpful for a better understanding: as table S1 with all samples and information of the Beishiku Temple analyzed in this study (this aspect of the paper results is not always clear and easy). I suggest a restructuring of the main figures and tables in the text to insert also this Supplementary table.
The list of corrections is reported as follows:
Line 70- Fungi hyphae and secreted oxalic acid are related to oxalates like calcium oxalate and whewellite (CaC2O4·H2O).
Why do you not also report the other form of calcium oxalate, the dehydrated form of Weddellite? (Salvadori, O.; Casanova Municchia, A. 2016). Moreover, in your results, you found this form of calcium oxalate.
Furthermore, fungi can synthesize CaCO3 with two crystal structures of either calcite or vaterite directly.
In literature, is well known the biomineralization of aragonite and also of monohydrocalcite. I suggest you to add all the carbonate calcium minerals formed by biomineralization for more complete and correct documentation of this phenomenon (Krumbein, 1974; 1979; Sánchez-Navas et al., 2009; Dhami, 2013. Ascaso, C., Wierzchos, J., Souza-egipsy, (2002).
In fig. 1 could you add some labels on the circle with the name of the area of sampling? ( EO-1/2; ET-1/2; SO1/2). This way results in a clearer to the reader of the different sampling positions.
Line 134
Are missed the information about the SEM/EDS measurement set-up ( voltage, filament current information, value of emission current, and the working distance ) for the SEM analysis (I suppose you have worked in SE mode) and for the EDS analysis (have you acquired the EDS spectra by the spot or area mode?).
Moreover, you don't describe anyone technique to avoid a collapse of the biological structures (e.g. chemical fixation with glutaraldehyde, dehydration in ethanol and a critical point dried) Normal 8/81: esame delle caratteristiche morfologiche al microscopio elettronico a scansione (SEM). Dornieden, et al.,(2000); Sterflinger, K. et al., (1997); Casanova Municchia, A. et al., (2018).
In fact, in your SEM images (Fig2a and b), all the biological structures appear completely collapsed losing the morphological information. If you don't use an environmental SEM (ESEM) you should prepare the biological samples following the method for biological sample preparation. Can you motivate me why you didn't make this process?
In line 388 le last letter is wrong (not l but J)
Line 435 - Lost the reference. The calcite crystal by bioricristillization can have different morphologies (Wei, S et al., (2015)).
In Figure 9f there is a peak before calcium, probably potassium. Why do you not label the chemical name on the spectrum? The same comment is for the line before the (gold) Au.
Line 508 - The value goes over the value of 100% for RH. Could be better to speak of saturation reached.
This study shows which microorganisms produce pigmentation or organic acid in the lab condition at room temperature. However, the production of pigments is straight dependent on the environmental conditions where stress factors such as high UV radiation, desiccation and hypersalinity occur. Hence, different behaviour could be found from lab analysis and the real condition. In the discussion, this aspect should be stressed.
Sakr, A. A., Ghaly, M. F., Edwards, H. G. M., Ali, M. F., & Abdel-Haliem, M. E. (2020). Involvement of Streptomyces in the deterioration of cultural heritage materials through biomineralization and bio-pigment production pathways: a review. Geomicrobiology journal, 37(7), 653-662.
Vítek, P., Edwards, H. G. M., Jehlicka, J., Ascaso, C., De los Ríos, a, Valea, S., … Wierzchos, J. (2010). Microbial colonization of halite from the hyper-arid Atacama Desert studied by Raman spectroscopy. Philosophical Transactions. Series A, Mathematical, Physical, and Engineering Sciences, 368(1922), 3205–3221. https://doi.org/10.1098/rsta.2010.0059
Line 642- In samples collected from various types of appearance and locations, as well as microenvironmental conditions, the compositions of culturable microbial communities varied greatly. Factors of temperature, RH, and pH value were closely associated with distributions of the microbial community and composition.
Checking the environmental temperature, relative humidity, and pH to map the variation and distribution of microbial communities is a correct and accurate approach. However, for this purpose, a greater sampling number of positions could give more accurate statistical data. You should stress this point in the discussion section.
Reviewer 3 Report
This is a useful study of biodeterioration of the sandstone Beishiku Temple by microorganisms (bacteria and fungi). The manuscript is interesting and well-structured. The role of cultivated microorganisms in the bioweathering of the stone heritage is shown in this paper. A potential role in the biomineralization of calcium oxalate and calcite under the action of fungi isolated from the surface of damaged sandstone is shown.
I have the following comments which I hope will be helpful to the authors
I would recommend more clearly presenting the data of XRD analysis in Figures 9c and 9f. Symbols of chemical elements are not visible on the spectrum.
And also in the caption to Figure 9, is it possible to indicate that Figures c and f represent the analysis data. This is not very clear right away when reading the caption to the figure, it becomes clear only in lines 594 and 595 in section 4.5.
I would recommend the authors to make Figure 2 larger. Maybe place photos as well as in Figure 9.
